The complete mitochondrial genome of Dysgonia stuposa (Lepidoptera: Erebidae) and phylogenetic relationships within Noctuoidea

Sun Yuxuan
Zhu Yeshu
Chen Chen
Zhu Qunshan
Zhu Qianqian
Zhou Yanyue
Zhou Xiaojun
Zhu Peijun
Li Jun healthlicn@chnu.edu.cn
Zhang Haijun haijunzhang@163.com
College of Life Sciences, Huaibei Normal University , Huaibei , Anhui , China
Zhang Jia-Yong
Electronic publication date: 2020 Mar 16
Publication date: 2020
Volume: 8
Electronic Location ID: e8780
Received 2019 Jul 26; Accepted 2020 Feb 21
Copyright: ©2020 Sun et al.
Copyright year: 2020
Copyright holder: Sun et al.
License: This is an open access article distributed under the terms of the Creative Commons Attribution License, which permits unrestricted use, distribution, reproduction and adaptation in any medium and for any purpose provided that it is properly attributed. For attribution, the original author(s), title, publication source (PeerJ) and either DOI or URL of the article must be cited.
License URL: https://creativecommons.org/licenses/by/4.0/

Keywords: Phylogenetic relationship, D. stuposa, Mitochondrial genome, Noctuoidea

Funding: National Undergraduate Training Program for Innovation and Entrepreneurship 201810373111 Youth Project of Anhui Natural Science Foundation 1908085QC94 Anhui Natural Science Foundation 1908085MC86 Key Program of Anhui Natural Science Research Projects KJ2019A0958 This work was supported by the National Undergraduate Training Program for Innovation and Entrepreneurship (Grant No. 201810373111), Youth Project of Anhui Natural Science Foundation (1908085QC94), general Project of Anhui Natural Science Foundation (1908085MC86) and Key Program of Anhui Natural Science Research Projects (KJ2019A0958). The funders had no role in study design, data collection and analysis, decision to publish, or preparation of the manuscript.

==============================
To determine the Dysgonia stuposa mitochondrial genome (mitogenome) structure and to clarify its phylogenetic position, the entire mitogenome of D. stuposa was sequenced and annotated. The D. stuposa mitogenome is 15,721 bp in size and contains 37 genes (protein-coding genes, transfer RNA genes, ribosomal RNA genes) usually found in lepidopteran mitogenomes. The newly sequenced mitogenome contained some common features reported in other Erebidae species, e.g., an A+T biased nucleotide composition and a non-canonical start codon for cox1 (CGA). Like other insect mitogenomes, the D. stuposa mitogenome had a conserved sequence ‘ATACTAA’ in an intergenic spacer between trnS2 and nad1, and a motif ‘ATAGA’ followed by a 20 bp poly-T stretch in the A+T rich region. Phylogenetic analyses supported D. stuposa as part of the Erebidae family and reconfirmed the monophyly of the subfamilies Arctiinae, Catocalinae and Lymantriinae within Erebidae.

Introduction

Dysgonia stuposa (Lepidoptera: Erebidae) is an important pest species, and it has a wide distribution throughout the southern and eastern parts of Asia. Its larvae mainly consume the leaves of Punica granatum (pomegranate) resulting in considerable economic losses. In the northern areas of China, D. stuposa pupates during the winter to avoid the harsh environment (Piao, Fan & Zheng, 2012). The identification and prevention of D. stuposa at the pupal stage based on morphological characteristics is quite difficult for taxonomists and population ecologists. Despite the economic importance, our understanding of D. stuposa biology or phylogenetic status at the molecular level is still in its infancy. New molecular techniques such as DNA barcoding and PCR-RFLP are considered more reliable than morphology for studying taxonomy of animals (Arimoto & Iwaizum, 2014; Raupach et al., 2010). The application of molecular techniques to study the sequence of D. stuposa mitogenome will help in its precise identification and classification while contributing to future genetic ecology and evolutionary analyses.

The insect mitogenome is typically a 14–19 kb sized, circular, double-stranded DNA molecule (Boore, 1999). Compared to the nuclear genome, mitogenome is small in size and comparatively easy to sequence. Mitogenome usually contains numerous typical characteristics, such as stable gene composition, and conserved gene arrangements, which are widely used in molecular identification, population genetics, systematics and biogeographic studies (Wolstenholme, 1992; Wilson et al., 2010). Given the vast diversity of insects, mitogenome analyses are beneficial for species identification and broadly employed in the study of genomic evolution and phylogenetic relationships (Lu et al., 2013; Cameron, 2014).

Noctuoidea is one of the largest superfamilies of Lepidoptera, with over 42,400 described species (Nieukerken et al., 2011). Unlike other superfamilies, a metathoracic tympanal organ is a characteristic feature of Noctuoidea species (Miller, 1991). However, morphological based phylogenetics has failed to resolve classification conflicts at the family and sub-family level. Furthermore, the initial molecular studies were also unable to provide sufficient information as most of them rely on one or two genes with only 29–49 species (Mitchell et al., 1997; Fang et al., 2000). Mitchell, Mitter & Regier (2006) conducted systemic analyses based on two nuclear genes (elongation factor-1α (EF-1α) and dopa decarboxylase (DDC)) and increased taxon sampling (146 species), that supported the monophyly of sub-families and proposed a LAQ clade (Lymantriidae and Arctiidae became subordinate subfamilies within quadrifid noctuids). Zahiri et al. (2011) reconstructed the molecular phylogenetics of Noctuoidea using one mitochondrial (cox1) and seven nuclear genes (EF-1α, wingless, RpS5, IDH, CMDH, GAPDH and CAD) from 152 species with the Maximum Likelihood (ML) method. They proposed a new perspective, splitting up the traditional group of quadrifid noctuids, and re-establishing Erebidae and Nolidae as families (Zahiri et al., 2011). However, this study failed to clarify phylogenetic relationships between Erebidae subfamilies (Zahiri et al., 2012). Additionally, morphological studies were not entirely consistent with the molecular studies in challenging some traditional synapomorphies, such as the “quadrifid” forewing venation and the presence of a transverse sclerite in the pleural region of segment A1 (Minet, Barbut & Lalanne-Cassou, 2012).

Complete mitogenomes and the mitochondrial genes are increasingly applied to understand phylogenetic relationships. For example, Wang et al. (2015) proposed two new tribes and established relationships between them within Lymantriinae by using two mitochondrial genes (cox1 and rrnL) along with six nuclear genes, using ML and Bayesian Inference (BI). The nucleotide and amino acid sequences of mitochondrial PCGs are also broadly used to determine the taxonomic status of species and to analyze phylogenetic relationships within Erebidae (Yang & Kong, 2016; Liu et al., 2017). Furthermore, as the mitogenome differs from the nuclear genome, it has been increasingly used to investigate poorly supported phylogenetic questions such as the position of Nymphalidae within Papilionoidea (Yang et al., 2009). Since many species of the genus Dysgonia have been moved to other genera, including Erebidae and Noctuidae based on the classification of Holloway & Miller (2003), the taxonomic status of many species remained uncertain. In our study, we sequenced the complete mitogenome of D. stuposa and reconstructed phylogenetic relationships to assess its phylogenetic position within Noctuoidea. The newly sequenced mitogenome supported new phylogenetic relationships within Erebidae and will provide a foundation for further studies into Noctuidae and Erebidae mitogenomics, biogeography, and phylogenetics.

Material and Methods

Specimen collection and Genomic DNA extraction

The D. stuposa moths were collected from Xiangshan mountains (N33°59′, E116°47′), Huaibei, Anhui, China. Based on morphological characteristics, the collected specimens were identified as D. stuposa using the record in Fauna Sinica (Chen, 2003). The genomic DNA (contains nuclear genome and mitogenome) of D. stuposa was isolated using the Animal Genomic DNA Isolation Kit according to the manufacturer’s instructions (Sangon, Shanghai, China).

PCR amplification and fragment sequencing

To amplify the D. stuposa mitogenome, the universal (F1-R13) and specific primers (S1F-S3R) were used to perform PCR amplification (Table 1) (Sun et al., 2016). All PCR amplifications were executed using high fidelity DNA Polymerase (PrimeSTAR® GXL, Takara, Dalian, China). PCRs was performed according to Sun et al. (2016) with extension times depending on the putative length of target fragment. PCR product size was determined by agarose gel with TAE buffer, then sequenced at General Biosystems (General, Chuzhou, China) in both forward and reverse directions using ABI 3500 Genetic Analyzer by the Sanger sequencing method. For long fragments, internal sequencing primers were designed based on known fragment sequence. For the A+T rich region, the fragment was sequenced from two directions and repeated three times.

Table 1 Details of the primers used to amplify the mitochondrial DNA of D. stuposa.

Primer name	Nucleotide sequence (5′–3′)	
F1	TAAAAATAAGCTAAATTTAAGCTT	
R1	TATTAAAATTGCAAATTTTAAGGA	
F2	AAACTAATAATCTTCAAAATTAT	
R2	AAAATAATTTGTTCTATTAAAG	
F3	ATTCTATATTTCTTGAAATATTAT	
R3	CATAAATTATAAATCTTAATCATA	
F4	TGAAAATGATAAGTAATTTATTT	
R4	AATATTAATGGAATTTAACCACTA	
F5	TAAGCTGCTAACTTAATTTTTAGT	
R5	CCTGTTTCAGCTTTAGTTCATTC	
F6	CCTAATTGTCTTAAAGTAGATAA	
R6	TGCTTATTCTTCTGTAGCTCATAT	
F7	TAATGTATAATCTTCGTCTATGTAA	
R7	ATCAATAATCTCCAAAATTATTAT	
F8	ACTTTAAAAACTTCAAAGAAAAA	
R8	TCATAATAAATTCCTCGTCCAATAT	
F9	GTAAATTATGGTTGATTAATTCG	
R9	TGATCTTCAAATTCTAATTATGC	
F10	CCGAAACTAACTCTCTCTCACCT	
R10	CTTACATGATCTGAGTTCAAACCG	
F11	CGTTCTAATAAAGTTAAATAAGCA	
R11	AATATGTACATATTGCCCGTCGCT	
F12	TCTAGAAACACTTTCCAGTACCTC	
R12	AATTTTAAATTATTAGGTGAAATT	
F13	TAATAGGGTATCTAATCCTAGTT	
R13	ACTTAATTTATCCTATCAGAATAA	
S1F	ACTTTAAAAACTTCAAAGAAAAA	
S1R	ACTTAATTTATCCTATCAGAATAA	
S2F	CGCAACTGCTGGCACAAA	
S2R	GAAGAGAAGTTTATAGTGGATGAGGTT	
S3F	TAAGCTGCTAACTTAATTTTTAGT	
S3R	GTAATAAATTCCTCGTCCAATAT	

Table 2 Details of the lepidopteran mitogenomes used in this study.

Family	Subfamily	Species	Size (bp)	GenBank No.	
Erebidae	Arctiinae	Spilarctia subcarnea	15,441	KT258909	
		Lemyra melli	15,418	KP307017	
		Hyphantria cunea	15,481	GU592049	
		Nyctemera arctata albofasciata	15,432	KM244681	
		Callimorpha dominula	15,496	KP973953	
		Aglaomorpha histrio	15,472	KY800518	
		Amata formosae	15,463	KC513737	
		Cyana sp. MT-2014	15,494	KM244679	
		Paraona staudingeri	15,427	KY827330	
		Vamuna virilis	15,417	KJ364659	
	Catocalinae	Grammodes geometrica	15,728	KY888135	
		Catocala sp. XY-2014	15,671	KJ432280	
		Dysgonia stuposa	15,721	This study	
	Herminiinae	Hydrillodes lentalis	15,570	MH013484	
	Aganainae	Asota plana lacteata	15,416	KJ173908	
	Hypeninae	Paragabara curvicornuta	15,532	KT362742	
	Lymantriinae	Gynaephora minora	15,801	KY688086	
		Gynaephora aureata	15,773	KJ507132	
		Lachana alpherakii	15,755	KJ957168	
		Gynaephora qumalaiensis	15,753	KJ507134	
		Euproctis similis	15,437	KT258910	
		Somena scintillans	15,410	MH051839	
Noctuidae	Noctuinae	Agrotis ipsilon	15,377	KF163965	
		Agrotis segetum	15,378	KC894725	
	Hadeninae	Mythimna separata	15,329	KM099034	
		Protegira songi	15,410	KY379907	
	Amphipyrinae	Sesamia inferens	15,413	JN039362	
		Spodoptera exigua	15,365	JX316220	
		Spodoptera litura	15,383	KF701043	
		Spodoptera frugiperda	15,365	KM362176	
	Heliothinae	Helicoverpa armigera	15,347	GU188273	
		Helicoverpa zea	15,343	KJ930516	
		Helicoverpa assulta	15,400	KT626655	
		Heliothis subflexa	15,323	KT598688	
	Plusiinae	Ctenoplusia agnata	15,261	KC414791	
		Ctenoplusia limbirena	15,306	KM244665	
Nolide	Chloephorinae	Gabala argentata	15,337	KJ410747	
	Risobinae	Risoba prominens	15,343	KJ396197	
Notodontidae	Thaumetopoeinae	Ochrogaster lunifer	15,593	AM946601	
	Phalerinae	Phalera flavescens	15,659	JF440342	
outgroup		Bombyx mori	15,664	AY048187	
outgroup		Antheraea pernyi	15,566	AY242996	

Sequence assembly and annotation

The complete mitogenome was assembled using the DNAMAN (https://www.lynnon.com/index.html). Sequence annotation (supplied in supplemental files) was performed by MITOS2 Web Server (http://mitos2.bioinf.uni-leipzig.de/index.py) and confirmed by BLAST to homologous sequences in NCBI (https://blast.ncbi.nlm.nih.gov/Blast.cgi). To determine PCG initiation and termination codons, sequences were aligned with other published Noctuoidea sequences using ClustalX 2.0 (Larkin et al., 2007). AT skew and GC skew values were calculated using the methods given by Perna & Kocher (1995). MEGA 5.0 software was used to analyze relative synonymous codon usage (RSCU) (Tamura et al., 2011). tRNA genes were determined by tRNAscan Search Server (http://lowelab.ucsc.edu/tRNAscan-SE/) and secondary structures inferred from folding into their canonical clover-leaf structures (Lowe & Eddy, 1997). rRNA genes were determined by MITOS2 Web Server and confirmed by BLAST with the homologous sequences in NCBI. Tandem Repeats Finder (http://tandem.bu.edu/trf/trf.html) was used to analyze non-coding regions for tandem repeats (Benson, 1999).

Phylogenetic analysis

To infer the phylogenetic relationships among Noctuoidea at superfamily level, concatenated nucleotide sequence alignments for PCGs from 42 species (Table 2) was performed. All of the sequences were downloaded from GenBank. The Saturnidae species Bombyx mori (AY048187) and Antheraea pernyi (AY242996) (Liu et al., 2008) were used as outgroups. Sequences were aligned using ClustalX 2.0 software (Larkin et al., 2007). ML and BI were used to reconstruct phylogenetic relationships. For the ML analysis, nucleotide sequences were partitioned and performed in IQ-TREE (http://iqtree.cibiv.univie.ac.at/) with the best-fit model GTR+F+I+G4 (Trifinopoulos et al., 2016), and the clade support was investigated with 1000 bootstrap replicates. For the BI analysis, the GTR model and Invgamma rate variation across sites were presented and performed with MrBayes 3.2.6 (Ronquist et al., 2012). One cold chain and three heated chains were run with the dataset for 10 million generations with the tree being sampled every 1,000 generations. After discarding the first 25% samples as burn-in, posterior probabilities were calculated. The phylogenetic trees were visualized in FigTree software (http://tree.bio.ed.ac.uk/software/figtree/).

Results and Discussion

Genome organization and composition

The D. stuposa mitogenome is a circular DNA molecule, which is 15,721 bp in length (accession number: MK262707) (Fig. 1). The size of the newly sequenced mitogenome is comparable to other Noctuoidea species, which range from 15,377 bp (Agrotis ipsilon) to 15,801 bp (Gynaephora minora) (Table 3). The variation in size is generally due to differences in the length of their non-coding regions (intergenic spacers and A+T rich region) (Lv, Li & Kong, 2018). Annotation found the typical 37 genes and a non-coding A+T rich region like most of the sequenced insect mitogenomes (Table 4). An A and T biased nucleotide composition is a characteristic feature of insect mitogenomes (Boore, 1999), and D. stuposa is no exception. Nucleotide composition of D. stuposa was highly biased towards using A and T (A = 39.98%, T = 40.38%, G = 7.5%, C = 12.14%) (Table 3); 80.36% total A+T content is comparable to previously sequenced lepidopterans (ranges from 77.84% in Ochrogaster lunifer to 81.49% in Gynaephora minora).

Figure 1 Map of the mitogenome of D. stuposa.

tRNA genes are labeled according to the IUPAC-IUB one-letter amino acids; cox1, cox2 and cox3 refer to the cytochrome c oxidase subunits; cob refers to cytochrome b; nad1-nad6 refer to NADH dehydrogenase components. The moth was photographed by the corresponding author Jun Li.

Table 3 Composition and skew in mitogenomes of Noctuoidea species.

Species	Size (bp)	A%	G%	T%	C%	A+T %	AT skew	GC skew	
Whole genome	
D. stuposa	15,721	39.98	7.5	40.38	12.14	80.36	−0.005	−0.236	
A. plana lacteata	15,416	40.08	7.49	40.26	12.16	80.34	−0.002	−0.238	
V.a virilis	15,417	40.18	7.56	40.22	12.05	80.4	0.000	−0.229	
G. minora	15,801	40.97	6.77	40.52	11.75	81.49	0.006	−0.269	
R. prominens	15,343	40.25	7.8	40.82	11.13	81.07	−0.007	−0.176	
O. lunifer	15,593	40.09	7.56	37.75	14.6	77.84	0.030	−0.318	
A. ipsilon	15,377	40.38	7.71	40.87	11.04	81.25	−0.006	−0.178	
PCGs	
D. stuposa	11,269	33.80	10.91	44.64	10.65	78.45	−0.138	0.012	
A. plana lacteata	11,211	33.87	10.92	44.76	10.45	78.63	−0.138	0.022	
V. virilis	11,203	33.14	11.16	45.43	10.27	78.57	−0.156	0.042	
G. minora	11,237	34.72	10.11	44.98	10.2	79.7	−0.129	−0.004	
R. prominens	11,216	33.64	10.57	46	9.8	79.64	−0.155	0.038	
O. lunifer	11,266	32.47	12.08	43.26	12.19	75.73	−0.142	−0.005	
A. ipsilon	11,211	34.24	10.64	45.56	9.55	79.8	−0.142	0.054	
A+T rich	
D. stuposa	406	43.6	2.46	48.77	5.17	92.37	−0.056	−0.355	
A. plana lacteata	328	46.04	1.22	48.48	4.27	94.52	−0.026	−0.556	
V. virilis	362	44.48	1.1	50.55	3.87	95.03	−0.064	−0.557	
G. minora	449	43.21	2.67	49.44	4.68	92.65	−0.067	−0.273	
R. prominens	342	44.15	2.34	49.42	4.09	93.57	−0.056	−0.272	
O. lunifer	319	44.51	1.57	48.9	5.02	93.41	−0.047	−0.524	
A. ipsilon	332	46.08	1.51	48.8	3.61	94.88	−0.029	−0.410	

Table 4 List of annotated mitochondrial genes of D. stuposa.

Gene name	Start	Stop	Strand	Length	Anti- codon	Start codon	End codon	Intergenic nucleotides	
trnM	1	68	J	68	CAT	/	/	2	
trnI	71	138	J	68	GAT	/	/	8	
trnQ	147	215	N	69	TTG	/	/	55	
nad2	271	1,284	J	1,014	/	ATT	TAA	−2	
trnW	1,283	1,350	J	68	TCA	/	/	−8	
trnC	1,343	1,409	N	67	GCA	/	/	22	
trnY	1,432	1,496	N	65	GTA	/	/	9	
cox1	1,506	3,041	J	1,536	/	CGA	TAA	−5	
trnL2	3,037	3,103	J	67	TAA	/	/	0	
cox2	3,104	3,820	J	717	/	ATA	TAA	−35	
trnK	3,786	3,856	J	71	CTT	/	/	0	
trnD	3,857	3,923	J	67	GTC	/	/	0	
atp8	3,924	4,085	J	162	/	ATC	TAA	−7	
atp6	4,079	4,756	J	678	/	ATG	TAA	27	
cox3	4,784	5,572	J	789	/	ATG	TAA	2	
trnG	5,575	5,640	J	66	TCC	/	/	0	
nad3	5,641	5,994	J	354	/	ATT	TAA	34	
trnA	6,029	6,085	J	57	TGC	/	/	105	
trnR	6,191	6,256	J	66	TCG	/	/	10	
trnN	6,267	6,332	J	66	GTT	/	/	8	
trnS1	6,341	6,406	J	66	GCT	/	/	32	
trnE	6,439	6,506	J	68	TTC	/	/	50	
trnF	6,557	6,624	N	68	GAA	/	/	−17	
nad5	6,608	8,368	N	1,761	/	ATT	TAA	−3	
trnH	8,366	8,433	N	68	GTG	/	/	0	
nad4	8,434	9,772	N	1,338	/	ATG	TA	42	
nad4l	9,815	10,102	N	288	/	ATG	TAA	14	
trnT	10,117	10,181	J	65	TGT	/	/	0	
trnP	10,182	10,246	N	65	TGG	/	/	7	
nad6	10,254	10,784	J	531	/	ATT	TAA	14	
cob	10,799	11,959	J	1,161	/	ATG	TAA	−2	
trnS2	11,958	12,025	J	68	TGA	/	/	22	
nad1	12,048	12,986	N	939	/	ATG	TAA	1	
trnL1	12,988	13,055	N	68	TAG	/	/	65	
rrnL	13,121	14,428	N	1,308	/	/	/	37	
trnV	14,466	14,533	N	68	TAC	/	/	0	
rrnS	14,534	15,315	N	782	/	/	/	0	
AT-rich region	15,316	15,721	/	406	/	/	/	/	

AT ((A-T)/(A+T)) and GC skew ((G-C)/(G+C)) were calculated for the J strand (majority) (Perna & Kocher, 1995); with negative AT skew (−0.005) and GC skew (−0.236), indicating the presence of more Ts than As, and Cs than Gs, respectively (Table 3). Negative AT skew has been reported in several other insect species such as Asota plana lacteata (−0.002), Risoba prominens (−0.007) and Agrotis ipsilon (−0.006).

Protein-coding genes and codon usage

PCGs identified from the D. stuposa mitogenome had a total length of 11,269 bp, accounting for 71.7% of the mitogenome. In insects, most PCGs are on the J strand (majority), while some of them reside on the N strand (minority) (Simon et al., 1994). In D. stuposa, nine of the thirteen PCGs (nad2, cox1, cox2, atp8, atp6, cox3, nad3, nad6 and cob) are encoded on the J-strand, while the remaining PCGs (nad5, nad4, nad4L and nad1) are on the N-strand. An ATN codon initiated all PCGs except cox1, which uses a CGA codon, as in most Lepidoptera (Table 4). The utilize of non-canonical initiation codons for cox1 is a common feature across insects (Liu et al., 2016; Dai et al., 2016).

To estimate codon usage among Noctuoidea species and to assess similarities and variations in codon usage and distribution, PCGs nucleotide sequences of seven Noctuoidea (belonging to four families: Erebidae, Noctuidae, Nolidae and Notodontidae) were compared (Fig. 2). In D. stuposa phenylalanine (Phe), asparagine (Asn), leucine (Leu), methionine (Met), tyrosine (Tyr) and isoleucine (Ile) were the most commonly used amino acids, while cysteine (Cys) was the most rarely utilized amino acid. Codon usage is similar across Noctuoidea. Furthermore, we used the codons per thousand (CDspT) metric to illustrate the codons distribution in different species (Dai et al., 2015) (Fig. 3). CDspT results exhibited similar trends across the Noctuoidea superfamily, with the maximum CDspT value observed for Asn and Ile.

Figure 2 Comparison of codon usage within the mitochondrial genome of members of the Noctuoidea.

Lowercase letters (a, b, c and d) above species names represent the family to which the species belongs (a: Erebidae, b: Nolide, c: Notodontidae, d: Noctuidae).

Figure 3 Codon distribution in members of the Noctuoidea.

(A) CDspT of Dysgonia stuposa. (B) CDspT of Asota plana lacteata. (C) CDspT of Vamuna virilis. (D) CDspT of Risoba prominens. (E) CDspT of Gynaephora minora. (F) CDspT of Ochrogaster lunifer. (G) CDspT of Agrotis ipsilon. CDspT, codons per thousand codons.

Relative Synonymous Codon Usage (RSCU) for Noctuoidea species is presented in Fig. 4. Codons usage within a given amino acid varied between species. All codons were found in D. stuposa, except ACG and CCG. Some noctuid species lack GC rich synonymous codons, with G or C at the third codon position, such as GCG, CGC, GGC and CCG (e.g., these are not present in A. ipsilon) (Wu, Cui & Wei, 2015). The rarity or complete absence of GC-rich codons occur in various insect species (Sun et al., 2017; Li et al., 2018).

Figure 4 Relative Synonymous Codon Usage (RSCU) of the mitochondrial genome of four families in the Noctuoidea.

(A) RSCU of Dysgonia stuposa. (B) RSCU of Asota plana lacteata. (C) RSCU of Vamuna virilis. (D) RSCU of Risoba prominens. (E) RSCU of Gynaephora minora. (F) RSCU of Ochrogaster lunifer. (G) RSCU of Agrotis ipsilon. (H) Codon families of synonymous codon. Codons indicated above the bar are not present in the mitogenome.

Figure 5 Predicted secondary structures of the 22 tRNA genes of the D. stuposa mitogenome.

(A–V) Twenty-two tRNA secondary structures.

Ribosomal RNA and transfer RNA genes

The D. stuposa mitogenome contains the large (rrnL) and small ribosomal genes (rrnS), encoded by the N strand with a length of 1,308 bp and 782 bp, respectively (Fig. 1, Table 4). In D. stuposa, rrnL was located between trnL1 and trnV, while rrnS was resided between trnV and the AT-rich region, as reported in previously sequenced mitogenomes (Yang et al., 2009).

There are 22 tRNA genes in the D. stuposa mitogenome, ranging in size from 57 bp (trnA) to 71 bp (trnK) (Table 4). Almost all tRNAs had the canonical clover-leaf secondary structure, except trnS1 that lacks the dihydrouridine (DHU) arm (Fig. 5), a common feature of trnS1 across mitogenomes of insects (Lavrov, Brown & Boore, 2000; Zhang et al., 2013). Stem pair mismatches in the secondary structure of tRNAs were observed such as an A-A mismatch (trnM), U-G mismatches (trn I, trnQ, trnW, trn Y, trnL2, trnG, trnF, trnH, trn T, trnP, trnV), U-U mismatches (trn Y, trnL2, trnS2) and a U-C mismatch (trnA). These mismatches may be corrected by an RNA-editing process which was proposed by Lavrov, Brown & Boore (2000), but has not been investigated fully in Lepidoptera.

Overlapping, intergenic spacer and A+T rich regions

Overlapping genes has been proposed to extend the genetic information possibly within the limited size of the genome, and are commonly observed in metazoan mitogenomes (Wolstenholme, 1992). We identified nine overlapping regions, a total length of 144 bp (Table 4). A seven bp overlapping region present at the boundary of atp6 and atp8 has also been reported in many other insects. The D. stuposa mitogenome also had 21 intergenic spacer regions, ranging in size from 1 to 105 bp. The 105 bp spacer located between trnA and trnR and had high A and T content (A = 47.62% and T = 49.52%) and a similar spacer has been described in Andraca theae (77 bp spacer with A = 46.75% and T = 44.16%). We also observed a 22 bp spacer that contained an ‘ATACTAA’ motif located between nad1 and trnS2 (Fig. 6A). This region commonly exists in most insect mitogenomes even though the region varies in size between lepidopteran species (Cameron & Whiting, 2008).

Figure 6 Features in the intergenic spacer and the A+T rich region.

(A) Alignment of the intergenic spacer region between trnS2 and nad1 of several Noctuoidea insects. (B) Features present in the A+T-rich region of D. stuposa. The ‘ATAGA’ motif is shaded. The poly-T stretch is underlined and the poly-A stretch is double underlined. The single microsatellite ‘AT’ repeat sequence is indicated by dotted underlining.

Figure 7 The phylogenetic relationships within Noctuoidea.

(A) Tree showing the phylogenetic relationships among 43 species, constructed using Maximum Likelihood with 1000 bootstrap replicates. (B) Tree constructed using Bayesian Inference (BI) MCMC consensus tree, with posterior probabilities shown at nodes. Bombyx mori (AY048187) and Antheraea pernyi (AY242996) were used as outgroups.

Metazoan mitogenomes usually have a single large non-coding region, named as the A+T rich region (Clayton, 1991). It contains initiation signals for DNA transcription and replication (Fernández-Silva, Enriquez & Montoya, 2003). The A+T rich region of D. stuposa mitogenome is located between rrnS and trnM and is 406 bp in size (Table 4), with the negative GC skew (−0.355) and highest A+T content (92.37%) of the genome (Table 3). The A+T rich region usually contains multiple tandem repeat elements (Zhang & Hewitt, 1997); however, D. stuposa did not have macro-repeats but does include short repeating sequences. It has the ‘ATAGA’ motif along with a 20 bp poly-T repeat, a microsatellite-like (AT)10 repeat and a poly-A repeat sequence upstream of trnM (Fig. 6B). The poly-T stretch varies between different species (Dai et al., 2015), but the ‘ATAGA’ motif is conserved in insects (Zhang & Hewitt, 1997).

Phylogenetic relationships

To determine the phylogenetic position of D. stuposa, we reconstructed phylogenetic relationships with Noctuoidea species. In phylogenetic analyses, mitogenome PCGs have a lower sensitivity to analytical bias compared to other genes such as the tRNA or rRNA genes (Yang et al., 2015). Here, we applied the nucleotide sequence of the 13 PCGs for phylogenetic analyses using BI and ML methods. Results showed that D. stuposa is closely related to Grammodes geometrica, a clade that was well supported by both the methods (Figs. 7A and 7B). D. stuposa belongs to the family Erebidae and subfamily Catocalinae, consistent with the reported classification of Erebidae (Zahiri et al., 2011). Erebidae is a large noctuid family (Yang et al., 2015); however, its monophyly remained unconfirmed, especially for Catocalinae (Zahiri et al., 2012). In the present study, the Catocalinae was found monophyletic, but nodal support values were not significant, i.e., 0.76 posterior probability (BI) and 31% bootstrap values (ML). There is still some controversy about relationships of Catocalinae under Erebidae. Zahiri et al. (2011) demoted Catocalinae to a tribe Catocalini within the subfamily Erebinae, and upgraded Anobini (formerly as a tribe within Catocalinae by Holloway (2005) to subfamily Anobinae. Several species of the Dysgonia genus have been reclassified into Noctuidae (Holloway & Miller, 2003), results in further complications for phylogenetic analysis. Within Erebidae, our study supported the monophyly of subfamilies and suggested that Catocalinae is a subfamily, most closely related to Hypeninae (BI) or Aganainae (ML) (Figs. 7A and 7B). Furthermore, Noctuoidea contained four families: Notodontidae, Erebidae, Nolidae and Noctuidae, for which their phylogenetic relationship was Notodontidae + (Erebidae + (Nolidae + Noctuidae)) with strong nodal support in both ML and BI trees. Since there is limited data of complete mitogenome sequences from Oenosandridae and Euteliidae in the public repository NCBI, our results are consistent with the previous family-level phylogenetic hypothesis proposed by Zahiri et al. (2011).

Supplemental Information

Supplemental Information 1 The annotation of D. stuposa mitogenome

Click here for additional data file.

Supplemental Information 2 The complete mitochondrial genome of Dysgonia stuposa

Click here for additional data file.

Additional Information and Declarations

Competing Interests

Author Contributions

Data Availability

The authors declare there are no competing interests.

Yuxuan Sun conceived and designed the experiments, performed the experiments, analyzed the data, prepared figures and/or tables, authored or reviewed drafts of the paper, and approved the final draft.

Yeshu Zhu and Haijun Zhang conceived and designed the experiments, performed the experiments, prepared figures and/or tables, authored or reviewed drafts of the paper, and approved the final draft.

Chen Chen performed the experiments, prepared figures and/or tables, authored or reviewed drafts of the paper, and approved the final draft.

Qunshan Zhu performed the experiments, prepared figures and/or tables, and approved the final draft.

Qianqian Zhu performed the experiments, analyzed the data, prepared figures and/or tables, and approved the final draft.

Yanyue Zhou, Xiaojun Zhou and Peijun Zhu analyzed the data, prepared figures and/or tables, and approved the final draft.

Jun Li conceived and designed the experiments, performed the experiments, authored or reviewed drafts of the paper, and approved the final draft.

The following information was supplied regarding data availability:

The complete mitochondrial genome sequence of Dysgonia stuposa is available in the Supplemental Files and at GenBank: MK262707.

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
