# Peer review of "The complete mitochondrial genome of Dysgonia stuposa (Lepidoptera: Erebidae) and phylogenetic relationships within Noctuoidea"

_PeerJ, doi:10.7717/peerj.8780_

## Round 0.1 · original submission · Major Revisions

I agree the second reviewers’ comments. The paper need revision in the introduction, methods, results and discussion sections. The English writing must be read by a fluent English speaker.

If the authors want to use one mt genome to discuss the phylogenetic relationship of Noctuidae and Erebidae, they need sequence more representative samples. I also suggest the authors focus on the relationship within Erebidae. All the mt genomes of Erebidae should be added to analyze in the phylogenetic discussion. Why did the authors choose seven species to comapre the RSCU...? I found some species not belonging to Erebidae. You should explain why did you choose 7 species? In the methods, the model seclection and partition selection methods must be added.

The phylogenetic analyses in ML should be used PhyML or RaxML or ML... because ML analyses in Mega 5.0 are not good choose. All model parameter should be listed.

I also think ten authors are too many people in discussing one mt genome paper. If it is possible, some minor authors should be listed in acknowledge.

The authors should be revised carefully according to the second reviewer's comments.

Reviewer 1 ·

Basic reporting

no comment

Experimental design

no comment

Validity of the findings

no comment

Additional comments

In this study, Sun et al., sequenced and analyzed a mitogenome of Dysgonia stuposa (Lepidoptera: Erebidae). Further, they compared the mitogenome features with other species and constructed the phylogenetic relationships among 29 species of Noctuoidea. Since the classification between Noctuidae and Erebidae under the superfamily Noctuoidea is still controversial and unclear, the precise classfication of this species will be very helpful in phylogenetic relationship analysis of Noctuoidea. However, there are still some things the authors can do to improve their manuscript.
1. Line 84, ‘Fauna Sinica’ should be italicized.
2. The sentence in line 231, ‘since, ……sharing apomorphies’ has grammatical errors, please revised.
3. Line 246, 247, the color of the period should be black, not red.
4. The key point of this study is the phylogenetic analyses, however, the author only used one species in Catocalinae subfamily. Till now, is there only one species whole mitogenomes has been published in this subfamily?
5. All the references need to unified in the same format, and the authors list need ‘et al’ when the authors number more than six. In addition, the journal’s name should be unified abbreviation or not.
6. Figure 5 need to label the stem and loop structure to make readers can understand the tRNA structure easily.

Reviewer 2 ·

Basic reporting

Unprofessional and ambiguous English has been used mostly. There are still many suggestions for improving language usage to clarify in introduction, results and discussion.
The raw sequencing data should be deposited at the NCBI (GenBank) and the accession numbers should be provided in the manuscript.
More details in methods section are needed. eg. Partition set, or model parameter set. I did not support the one mitochondrial genome of D. stuposa can give a convincing result in discussion of the relationship of Noctuidae and Erebidae. If the authors want to discuss the phylogenetic relationship between Noctuidae and Erebidae, they should sequence more samples of Families Noctuidae and Erebidae. But they only sequenced one mt genomes from Erebidae. So I did not think this question should be discussed in the paper. If the authors really want to discuss this relationship, they need sequence more mt genomes from Noctuidae and Erebidae. And they should introduce the detail views from all researchers using the different analyses methods and different molecular markers, and so in the section of introduction.
I also found the authors did not use all mt genomes from GenBank, eg. Amata formosae KC513737, Cyana sp. MT-2014 KM244679 , Hydrillodes lentalis MH013484, Spilarctia subcarnea KT258909, Spilarctia alba KX753670. Why did the authors delete those mt genomes in their analyses methods? They should give an explanation. The figure of BI tree should be consist with the figure ML tree in colour. In a word, the authors should reanalyze the data and rewrite the results and discussion.

Experimental design

Methods have not been described in sufficient detail. It is not clearly specified how was the long PCR fragments sequenced. Used specific primers? Or NGS? Especially in AT-rich region, how can they obtain the high TTTTTTT…. Region? How can the author identify the sequenced high TTT..region right? I failed to find the method of sequencing in the fragment sequencing.
It is not clear how did the authors separate the mitochondria genome from the nuclear genome. Were the mitochondria physically isolated / purified before DNA extraction, or was the complete picture of nuclear and mtDNA sequenced together. How were the mitochondrial reads separated from those from nuclear genome ? These are essential details required in the methods and results section
The listed primers can not obtained the whole mt genomes! Please check the primers.

Validity of the findings

The conclusions that the authors draw from the assembled and annotated genome are valid. However, as discussed above, more details on methods used for sequencing of the assembled genome are essential to evaluate the quality of the assembled genome sequence

Additional comments

The authors have sequenced, assembled and annotated the mitochondrial genome of D. stuposa. The authors have then generated a phylogenetic tree based on 13 protein coding genes from mitochondria, covering some additional lepidopteran genomes. The results are mostly descriptive and have been are supported by their analysis. However, a few key details are missing from the methods section, and these must be included for a proper evaluation of the complete analysis.

Major comments:
1. The raw data must be submitted to the NCBI or an equivalent public database.
2. More details in methods section are needed. eg. Partition set, or model parameter set. I did not support the one mitochondrial genome of D. stuposa can give a convincing result in discussion of the relationship of Noctuidae and Erebidae. If the authors want to discuss the phylogenetic relationship between Noctuidae and Erebidae, they should sequence more samples of Families Noctuidae and Erebidae. But they only sequenced one mt genomes from Erebidae. So I did not think this question should be discussed in the paper. If the authors really want to discuss this relationship, they need sequence more mt genomes from Noctuidae and Erebidae. And they should introduce the detail views from all researchers using the different analyses methods and different molecular markers, and so in the section of introduction.
I suggest the authors added more mt genomes from GenBank and reanalyze the BI and ML trees to discuss the relationship within subfamilies of Erebidae not relationship between Noctuidae and Erebidae. So some background of the relationship of within subfamilies of Erebidae should be added in the introduction.
Unprofessional and ambiguous English has been used mostly. There are still a few suggestions for improving language usage to clarify in introduction, results and discussion.
The raw sequencing data should be deposited at the NCBI (GenBank) and the accession numbers should be provided in the manuscript.
More details in methods section are needed. eg. Partition set, or model parameter set. I did not support the one mitochondrial genome of D. stuposa can give a convincing result in discussion of the relationship of Noctuidae and Erebidae. If the authors want to discuss the phylogenetic relationship between Noctuidae and Erebidae, they should sequence more samples of Families Noctuidae and Erebidae. But they only sequenced one mt genomes from Erebidae. So I did not think this question should be discussed in the paper. If the authors really want to discuss this relationship, they need sequence more mt genomes from Noctuidae and Erebidae. And they should introduce the detail views from all researchers using the different analyses methods and different molecular markers, and so in the section of introduction.
I also found the authors did not use all mt genomes from GenBank, eg. Amata formosae KC513737, Cyana sp. MT-2014 KM244679 , Hydrillodes lentalis MH013484, Spilarctia subcarnea KT258909, Spilarctia alba KX753670. Why did the authors delete those mt genomes in their analyses methods? They should give an explanation. The figure of BI tree should be consist with the figure ML tree in colour. In a word, the authors should reanalyze the data and rewrite the results and discussion.
3. The correct citation for the analysis softwares in the text should be added. Eg. DNAMAN, Blast, ClustalX, Mega 5.0, MrBayes3.2, and FigTree…
The references should be correctly cited in any places. Eg.
4. I failed to find the annotated mt mitochondrial genome, so I can not check the gene structure. Please submit the sequence in supplement to further check.
5. Line 116-119: Did all RNA can be identified by tRNAscsan-SE?
6. There are many blank in the text. Please check.
7. The English writing should be checked by English native.
8. The sentences are belonging to discussion should be moved from the section of results.
9. The gene writing should be consist in text, figure and table.
10. In reference, the species name should be italic. The journal name should be full name listed.
11. The listed primers can not obtain the whole mt genome.
12. Ten authors did one mt genomes. I suggest the paper should list really contribution authors not ten authors!

Minor revisions are marked yellow in PDF.

Annotated reviews are not available for download in order to protect the identity of reviewers who chose to remain anonymous.

---

## Round 0.2 · Major Revisions

One outstanding reviewer gave you many excellent suggestions.
I thought it will be helpful for your paper.
Please restructure and rewrite the results, discussion and conclusion.

Reviewer 1 ·

Basic reporting

no comment

Experimental design

no comment

Validity of the findings

no comment

Additional comments

The authors addressed all the questions and comments and significantly improved the manuscripts with appropriate corrections.

Reviewer 2 ·

Basic reporting

Although the paper is improved in some sections, the result, discussion and conclusion should be rewritten again. I did not think the result and discussion are good structure. There are many sentences of results existed in discussion. If the authors did not know how to deal with discussion, I suggest they can put results and discussion together.
Many English grammars should be improved. The native English people need to read the paper.

Experimental design

OK

Validity of the findings

As the first review.

Additional comments

Although the paper is improved in some sections, the result, discussion and conclusion should be rewritten again. I did not think the result and discussion are good structure. There are many sentences of results existed in discussion. If the authors did not know how to deal with discussion, I suggest they can put results and discussion together.
Many English grammars should be improved. The native English people need to read the paper.

Reviewer 3 ·

Basic reporting

The written English is not very good, I've attached an edited version of the manuscript with suggested changes to improve clarity.

Referencing is not up to standard for current publications unfortunately. In some instances very general statements are supported by very supported by citation of papers which made similar statements but weren't the first to propose such a hypothesis or to observe such an aspect of mitochondrial genomics. For example, you cite Junqueira for skew statistics, but they simply used skew as a statistic in their paper much like you do here. Perna & Kocher 1995 proposed this statistic, they are the ones who should be cited. Use of review papers or the original literature is needed for these instances. In other instances papers do not support the statements for which they are cited, either because the paper has been miscited or because the statement being made is not supported by any literature in this field. Most such instances I flagged in the edited manuscript, but I would encourage the authors to think hard about whether the best reference is being cited each time.

In a related vein, the text too often includes vague and/or general statements to the effect that variation or similarity exists, but without detailing how or in what way. Again I've flagged the more obvious ones in the edited copy, but I would strongly discourage the authors from making statements like "and other morphological characteristics are used for classification" (line 60). Either elaborate on these details or don't comment on them, not every aspect of the paper needs to be mentioned, but that which is mentioned needs to be developed into a detailed text such that the reader can learn about this system, species or genome from the paper, not just that features exist.

Structure has some positive aspects but most of the Discussion is just repeating the results, while the Conclusion section is repeating them for a third time. Discussion sections are about the importance and context of the results found. They are a separate section because they have a quite different purpose from the Results. The amount of repetition is marked on the manuscript - not much Discussion is left. I don't know if this journal allows the combined "Results and Discussion" section that many journals do, but many authors find this conceptually easier to write for than separate sections. Your editor can advise but I would explore restructuring the paper in this way.

Hypothetical scope of the paper is OK.

Experimental design

Research question is kind of loose but common across these types of papers.

There are two main problems with how the analyses are chosen for the study.

1). Genome analysis features.
I know that it is common to report statistics about mitogenomes in papers such as this - nucleotide composition, skew, RSCU etc. But what is often missing, and is the case here is why these features are being analysed? Particularly if the genome is kind of average - as is the case here - with statistics that fall within previously reported ranges or genomes. It is one thing if a genome is extreme - the most or least biased or skewed or whatever, as outlier species are useful to identify and to look at why they might be so biased. But reporting that 'its pretty well like everything else' is kind of pointless. All of these statistics can be used to inform other analyses - biases of these types affect phylogenetic analyses in some instances. But why did you do these analyses? What is actually being tested? What do you want to know that an RSCU statistic will tell you?
In contrast the case for doing the phylogenetic analyses is clearer - you want to know if this species is classified in the right family, what it is related to etc. No such justification for doing the statistics is offered, and just because a lot of other papers don't justify doing these statistics doesn't make it right (I routinely make this point in reviews, but clearly can only review so many papers per year). Please think about why you do the analyses which you do?

2). Phylogenetic analyses
Even though the questions are justified, the methods aren't of current high standard for a number of reasons.
First, analysing the amino-acid sequence rather than DNA is not best practice. Each time this has been examined in detail it has been shown that the amino-acid dataset has less resolving power than the DNA one. The issue with third codon positions is readily accounted for by excluding third codons from the analysis, or by using partitioning to accommodate the different mutational dynamics on third positions.
Secondly, the methods are somewhat confusing. No input partitions are listed for PartitionFinder (it groups between predetermined starting partitions, you need to list what they are) and the output partitions are not reported. "Performed in IQ-Tree" also doesn't make sense - they are separate pieces of software. IQ-Tree includes a partition and model finding algorithm (ModelFinder) but it isn't a shell for implementing PartitionFinder, so which software did you actually use for models/partitions? Why is RAxML used for ML tree inference if you are using IQ-Tree for partition/model inference - IQ-Tree is a perfectly good ML inference software, it seems inefficient to switch programs. One best-fit model is listed (mtMET+F+R5) but how is that if you did partitioning? Multiple models are more likely, or in the unlikely event that the same general model was preferred for all partitions, say so and also state whether you used those models as linked or unlinked in the analysis. For BI a 'mixed model' is listed (line 145) but that just means different partitions can have different models - which ones were actually used and what is the partitioning system? The number of generations in the BI run was far, far too few - 10 million would be a minimum for an analysis of this size.
Accordingly, the phylogenetic analyses needs to be completely repeated. Use DNA, use partitioning to compensate for third codon effects, get your software use sorted out so it is clear what is being used and why, report on partitioning choices and model outcomes, run the BI for much much longer.

Validity of the findings

1) Phylogenetic results suspect due to poor methods choice as outlined above, will need to be fixed before final judgement.

2) Annotation of the genome includes at least two obvious errors. rrnS and rrnL are on the N strand in most insects and all Lepidoptera sequenced to date. Either they are on N in this species and this was misreported on line 193, or this is the most divergent moth mitogenome yet sequenced in which case you can't state that it is comparable to other species. Secondly, an overlap of 65bp between trnH and nad4 is not going to be accurate. Insect mitogenomes do not overlap by these margins, but MITOS is not always accurate in reporting annotations. With respect, I don't believe that you have accurately compared the start codon for nad4 against other Noctuoid mitogenomes as stated in lines 122-123. If the 20 amino acids that make up this overlap align well to the first 20 amino acids of nad4 in other species please provide a figure illustrating the alignment? But I strongly suspect that this is an error.

3) The interpretation of some relationship between how 'advanced' an insect is (itself a very flawed concept - see Crisp & Cook 2005) and nucleotide composition is very flawed. Parafronus may have the lowest A+T% of any pterygote, but this ignores results from the other 20+ mayfly mitogenomes which are available. Parafronurus is not representative of mayflies, and so this primitive = low A+T argument is demonstrably wrong. Conversely, Gotsek (2010) never claim that Solenopsis (an ant) is the most advanced insect, or that there is any relationship between that and A+T%. No published study makes the claim made by the authors in line 243 and the data presented in this study doesn't support making it here. Remove all of this.

Additional comments

This paper has some potential but it needs to be extensively reanalysed and rewritten. There are major flaws in the use of citations and in the interpretation of previous publications, I would encourage the authors to deeply read in this field to correct these problems. Finally, the authors should focus on why this species was studied and what is unique findings of this study, not repeating analyses found in other similar papers that lack a purpose in the current one.

Annotated reviews are not available for download in order to protect the identity of reviewers who chose to remain anonymous.

---

## Round 0.3 · Minor Revisions

One reviewer gave many excellent suggestions. Please revise the paper according to the suggestions.

Reviewer 2 ·

Basic reporting

The paper improved greatly. I still suggestion the paper should be read by a English native person or organization.

Experimental design

The paper improved greatly.

Validity of the findings

All the data are good.

Additional comments

I still suggestion the paper should be read by a English native person or organization.

Reviewer 3 ·

Basic reporting

Greatly improved from the previous version, I've made more editorial suggestions to improve the written expression on a scanned-copy of the manuscript.

Experimental design

Again greatly improved. The phylogenetic portion now conforms to current standards and is repeatable based on the methods described. It is sufficiently rigorous for the questions posed and the discussion of the results.

One thing to note though in terms of interpreting the phylogenies so produced. Nodal support values are statistical measures of support for the hypothesis summarised by the tree. So a branch can be present such as the one for Catocalinae (mentioned in line 227) but have statistically significant support - as was the case for the node supports quoted. For Bootstrapping any values <70% are not significant, whereas posterior probabilities are more like p-values, anything below 0.9 is not significant. I've rewritten the sentence to reflect that, but for the future consider the significance of a nodal support, not just whether a branch is present or not.

Validity of the findings

Concerns expressed in the former review have been accounted for. The data presented is sufficient for the conclusions drawn.

Annotated reviews are not available for download in order to protect the identity of reviewers who chose to remain anonymous.

---

## Round 0.4 · accepted · Accept

The current version of the manuscript has been significantly improved. The authors provided better explanations of methods used in the study, and corrected some unclear or not scientifically sound sentences in the English grammar.

I think that the work can be now accepted for publication (in this version I noticed few typos throughout the text, but I guess that the authors still have the chance to correct them). eg. P3L37 D. should be corrected as "Dysgonia"; P6L93-94 the symbol should be corrected.